# Meristematic Connectome: A Cellular Coordinator of Plant Responses to Environmental Signals?

**DOI:** 10.3390/cells10102544

**Published:** 2021-09-26

**Authors:** Donato Chiatante, Antonio Montagnoli, Dalila Trupiano, Gabriella Sferra, John Bryant, Thomas L. Rost, Gabriella S. Scippa

**Affiliations:** 1Department of Biotechnology and Life Science, University of Insubria, Via Dunant, 3, 21100 Varese, Italy; donato.chiatante@uninsubria.it; 2Department of Biosciences and Territory, University of Molise, Contrada Fonte Lappone, 86090 Pesche, Italy; dalila.trupiano@unimol.it (D.T.); gabriella.sferra@unimol.it (G.S.); scippa@unimol.it (G.S.S.); 3Department of Bioscience, College of Life and Environmental Sciences, University of Exeter, Stocker Road, Exeter EX4 4QD, UK; j.a.bryant@exeter.ac.uk; 4Department of Plant Biology, College of Biological Sciences, University of California, One Shields Avenue, Davis, CA 95616, USA; tlrost@ucdavis.edu

**Keywords:** *Populus nigra* L., *Arabidopsis thaliana* L., meristems, connectome, vascular cambium, root apical meristem, shoot apical meristem, root procambial bundles

## Abstract

Mechanical stress in tree roots induces the production of reaction wood (RW) and the formation of new branch roots, both functioning to avoid anchorage failure and limb damage. The vascular cambium (VC) is the factor responsible for the onset of these responses as shown by their occurrence when all primary tissues and the root tips are removed. The data presented confirm that the VC is able to evaluate both the direction and magnitude of the mechanical forces experienced before coordinating the most fitting responses along the root axis whenever and wherever these are necessary. The coordination of these responses requires intense crosstalk between meristematic cells of the VC which may be very distant from the place where the mechanical stress is first detected. Signaling could be facilitated through plasmodesmata between meristematic cells. The mechanism of RW production also seems to be well conserved in the stem and this fact suggests that the VC could behave as a single structure spread along the plant body axis as a means to control the relationship between the plant and its environment. The observation that there are numerous morphological and functional similarities between different meristems and that some important regulatory mechanisms of meristem activity, such as homeostasis, are common to several meristems, supports the hypothesis that not only the VC but all apical, primary and secondary meristems present in the plant body behave as a single interconnected structure. We propose to name this structure “meristematic connectome” given the possibility that the sequence of meristems from root apex to shoot apex could represent a pluricellular network that facilitates long-distance signaling in the plant body. The possibility that the “meristematic connectome” could act as a single structure active in adjusting the plant body to its surrounding environment throughout the life of a plant is now proposed.

## 1. Introduction

Plants have the ability to coordinate a complex interaction between physiological, cytological, and molecular events, which is necessary for responding continuously to environmental signals [1,2,3,4,5,6]. Several recent advances in cell and molecular biology support the hypothesis that the root apex may act as the “brain” of the plant, being the centre of control for interaction coordination [7]. This concept was first proposed by Charles and Francis Darwin in 1880 in the *Power of Movement in Plants*, but later other authors [8,9,10] proposed that the transition zone (TZ) of the root plays the role of control for interaction coordination in plants. Support for their hypothesis comes from physiological and cytological properties of the TZ including: (a) ion flux oscillation and other specific transport processes related with oxygen and auxin; (b) oscillating electric spike activities [10,11]; (c) endocytosis-driven vescicle recycling [12,13,14]; (d) high oxygen demands [15].

However, several experiments performed by our research group [16,17,18,19,20] have demonstrated that roots respond to mechanical stresses through the action of the vascular cambium (VC) even when they lack all root primary tissues, including the root apical meristem (RAM) and the TZ. This fact casts a serious shadow over the proposal that TZ could play a role of “brain” of the plant. Moreover, evidence that will be reported and discussed below, indicates that responses to mechanical stresses involve similar VC activities in different zones of both root and stem organs. This evidence led us to hypothesize that these responses arise from an intense crosstalk between different portions of the VC probably involving signal transduction factors. 

A recent comparative translatome analysis [21] shows that, across plant species, mechanisms regulating meristematic cell activity are better conserved than those characterizing other cell populations. This supports our observations regarding the similarity of VC behaviour along the root-stem axis. For this reason, it is not unreasonable to suggest that homologous regulatory mechanisms could be active along the sequence of meristems that in a plant are organized (according to a bottom-top direction) to form (i) RAM, (ii) root procambial bundles (root PRC), (iii) root and shoot VC, (iv) shoot procambial bundles (shoot PRC), and (v) shoot apical meristem (SAM). If this is true, then the sequence of all these meristems could behave as a single functional unit that could be named “meristematic connectome”. However, it is important to highlight that there are considerable cytological, physiological, biochemical and functional differences between the “meristematic connectome” proposed here and a neural connectome reported in the literature on animal systems. The “meristematic connectome” could function as a cellular network for rapid communication through the distant plant body’s compartments while also being involved in responses to environmental signals.

## 2. The Loss of RAM and TZ Does Not Affect Plant Growth and Response under Mechanical Stresses

### 2.1. Pruning and Bending Treatments

During the past two decades our research group has worked to understand how trees respond to external signals that threaten their anchorage to the ground. In particular, our studies aimed to understand whether a modification of root architecture can take place at any stage of the tree development and if new lateral roots (LRs) can be produced even when parental roots lack primary tissues. For this reason, in all our experiments the starting plant material was a pruned root system from which all primary tissues had been removed. 

In seedlings of *Pinus*, *Fraxinus* and *Populus* the pruning treatments did not affect root growth potential (RGP) as shown by the emergence of new LRs from woody parental roots. This led to the increase of root biomass and length [18,22,23] (Figure 1) and demonstrated that this response is conserved in both gymnosperm and angiosperm plant species.

In the second series of experiments with *Fraxinus* and *Populus*, we have investigated the response to mechanical stress, induced by the application of bending, along the root axis of a plant lacking all primary tissues. In particular, the woody taproot axis was divided into three sectors named respectively (i) Above Bending Sector (ABS), Below Bending Sector (BBS), and Bending Sector (BS). The published in silico model [24] showed that, within each sector, tension forces are active on the convex side whereas compression forces occur on the opposite concave side. Moreover, a more recent and newly developed model [25] indicated that the magnitude of both tension and compression forces dissipate rapidly and symmetrically moving from BS toward the ABS and BBS. The data from the bending treatment showed that plants remain viable, responding to mechanical stress through a unidirectional formation of reaction wood (RW) and new LRs. In particular, we observed the formation of RW only toward the concave side of the BS (Figure 2A), which was characterized by a low number of vessels as well as mechanical fibres with poorly lignified cell walls, a high carbohydrate content and a gelatinous layer [25]. Furthermore, in the concave side of the BS we found the highest number of cambial cells, which was similar to non-bent roots (control) in the convex side and in both sides of ABS and BBS (Figure 2B). 

Finally, the LRs formed only toward the convex side of the taproot (Figure 3), with the highest concentration in both the BS and ABS, in accordance with data reported in other studies for *Arabidopsis thaliana* bent root [26,27,28].

A comparable response of VC was also observed in poplar bent stem where RW is formed in the convex side of ABS [25] in contrast to what was previously found in the roots where is formed in the concave side [29]. Thus, as shown in the model proposed by De Zio et al., [25], despite the symmetrical force dissipation toward the two opposite sides of the bending sector, in stem only the convex side produces RW and in root only the convex side of ABS produces new LRs, while the concave side of BBS produces the RW. 

The differences observed along the bent taproot and between bent root and stem, highlighted that the VC can modulate its activity to respond differently “when and where” it is necessary.

A possible interpretation of this response specificity is that the plant is able to apply a “priority criterion” after a “costs-benefits” balance assessment, which resulted in costs involved in the construction of higher wood tissues area related to the benefit of a major water transport in the stem convex side and root concave side. Other examples of cost-benefit balance assessment that lead to a phenotype modification are in response to (i) quantitative and qualitative variation in the light environment [30,31,32], (ii) soil water shortage with increasing the number of very fine roots to enhance water uptake [20,33,34], and (iii) higher soil water availability which leads to an increase of fine root diameter to enhance water transportation [35]. 

Moreover, the occurrence in plants of an internal competition for resources between different root and shoot branches is another demonstration that plants are normally able to assess gains and losses before making a decision that provides the best profit outcome [36,37]. Adaptation of body architecture to the living environment is a further typical example of how plant response can be based upon the assessment of environmental signals followed by an “adaptive” decision [38,39]. 

### 2.2. Detection and Transduction of Mechanical Stresses

The difference of responses along the root and shoot axis, observed following our bending treatments, raises the question of how the VC assesses (a) direction and magnitude of mechanical forces and (b) how the signal is later transduced to induce the appropriate responses in sites of VC, even though they may be distant from where bending stress is applied. 

To respond to this question, it is necessary to consider first what is known in the literature regarding mechanical stress perception and thus its effects on (i) plasma membranes, (ii) microtubules (MT) and (iii) actin filaments. In fact, the plasma membrane is comparable to a fluid and therefore it is affected only by isotropic forces [40,41], whereas actin filaments and MTs are better suited to sense the direction of mechanical forces due to their stiffer and more extended nature [42]. It has been experimentally proved that MTs subjected to tension-stress align along the tension lines whereas aligning orthogonally to compression lines with a random alignment after the release of mechanical stress [43]. Among other signals, auxin can influence microtubule organization and orientation during auxin-dependent growth changes and under mechanical stress conditions [44]. As the interaction between cortical-MTs and auxin is not direct, it was proposed that mechanical stress could act as the common input controlling both cortical-MT orientation and PIN1 polarity in *Arabidopsis* [45].

Moreover, mechanical forces can influence plasma membranes, microtubules, actin filaments and calcium channels (CaC) and further, it is known that tension force can affect the function of cyclic nucleotide gated channels (CNGCs) and mechanosensitive calcium channels (MCA) through a membrane thinning effect [46,47,48,49,50].

Data obtained by us through a proteomic approach, highlight the involvement of a high number of functional proteins such as annexin, ankyrin, nucleotide diphosphate kinase (NDPK), phosphodiesterase, peroxidase, ara4-interacting protein, ROS signaling, F-actin binding and Ca^2+^ channel activities. The latter are the endomembrane-associated proteins responsible for transducing the signal and influence the asymmetrical VC response of the convex-stretched and concave-compressed side [29,51,52]. 

Furthermore, we found auxin strictly associated with the induction of VC activity and the unidirectional formation of RW toward the concave compressed side [16,17,25,29] whereas, according to Richter et al. [53], it preceded lateral root formation on the convex side of the curve. In other investigations, it has been observed that: (a) LRs may also form in the bent regions in the decapped root, indicating that mechanical forces can induce LRs formation in the absence of a gravitropic stimulation [53]; (b) *Arabidopsis* mutants for auxin transport or signaling show wild-type bending-related LR formation [26,53,54]. Besides auxin, we have found, in accord with Waidmann et al., [55] and Waidmann and Kleine-Vehn [56], that cytokinins (CK), in particular Z-type, act as central factors opposing gravitropism [25]. Furthermore, an antagonistic interaction of CKs and IAA, with opposite trends in bent stem and root seems to regulate organ-specific responses to mechanical constraints. In stems, the CK free bases could have a key role in the control of unidirectional RW formation, whereas the IAA could be specifically and asymmetrically accumulated only in the cambium zone to induce an earlier and more rapid RW production than in the bent root. Conversely, in root, a key role of IAA in the promotion of cambial cell division and RW initiation was confirmed [25].

In addition to hormones (presented above), we cannot overlook the possible involvement as mechanical stress signal transducers of mRNA, siRNAs, proteins, peptides, and neurotransmitter-like molecules [57,58,59,60,61]. 

Taking into account all these findings, according to Landrein and Hamant [62], we can speculate that mechanical stress direction and magnitude, changing anisotropic conformation in cell wall and membrane tension status, could lead the selective opening of a membrane-associated protein and the induction of signaling intermediates such as kinases, calcium or small GTPases that ultimately act on gene expression and trigger the relocation of actin filaments and impact on cortical-MT dynamics and/or organization. Among other signals, auxin distribution and CK/auxin ratio, could strongly influence cortical-MT organization and orientation. In this scenario, in the convex site, tension could promote actin polymerization and MTs alignment in tensile direction, able to increase cell wall resistance through the synthesis of cellulose microfibrils in the maximal direction of tensile stress [43]. Conversely, in the concave site, the compression forces could promote actin branching and change in MT orientation toward the new maximal tensile stress direction [40,41]. 

How can cortical-MTs discriminate between tension and compression for their alignment? This is one of the most difficult question to address here. However, the intrinsic structure of the microtubules and in particular their ability to withstand tension, while being destabilized by compression, together with their elongated-anisotropic shape, could be sufficient to make them tension or compression sensors on their own. Besides ensuring biomechanical functions, MT dynamics and/or organization in the VC could impact on the observed spatially-related strategies to maintain water uptake and transport in a deforming condition: increasing xylem thickness thought reaction wood formation toward the concave side of maximum point of bending (BS) and enhancing lateral root formation toward the convex side of BS and above maximum point of bending (ABS). 

In summary, the data presented above suggest that the VC responds to mechanical signals whenever and wherever needed; more importantly, the data also show clearly that the responses to mechanical stress follow the assessment of both stress direction and magnitude, and can be formed distant from the place where the mechanical stress is applied (Figure 4). These decisional activities by the VC suggest occurrence of an intense crosstalk between VC initials to coordinate and select the most fitting response to the stress signals. Therefore, it is not unreasonable to assume that, in responding to stress signals, the VC of a plant behaves as a single decisional structure.

If this is true, then an interesting scenario emerges according to which all meristems act as a unique structure when regulating the plant-environment relationship. We present below the hypothesis that there is a unique structure formed by the sequence of all meristems—RAM, root PRC, root VC, shoot VC, shoot PRC, SAM—that we call “meristematic connectome”. A strong support for the hypothesis of an intense crosstalk occurring within the “meristematic connectome” is represented by the fact that all meristematic cells can communicate through their plasmodesmata, thus acting as a pluricellular structure. This hypothesis is further supported by recent findings presented by Kutsher et al. [63] specifically regarding the spreading of virus in leaf cells throughout plasmodesmata.

## 3. Meristematic Connectome and the Coordination of Plant Response to Mechanical Stresses

A further support for the concept of a “meristematic connectome” formed by the sequence RAM, root PRC, root VC, shoot VC, shoot PRC, SAM, derives from the the recent demonstration that, beside RAM and SAM, PRC and VC also exhibit homeostasis (i.e., the pluripotent property of meristematic cells) [64,65]. In fact, Ojolo et al., [65] suggest that the regulatory mechanisms governing homeostasis are conserved by transferring this property to one daughter cell after each single cell division (probably through an epigenetic setup).

In plants that lack a secondary meristem (i.e., VC) as in the case of the majority of monocots, the “meristematic connectome” can still be active even though not involved in secondary growth. This may also be in those non-conventional monocots such as those in order *Asparagales* that possess an unusual lateral meristem (i.e., not homologous with the VC) but which is nevertheless responsible for secondary growth [66]. We highlight below a number of morphological and functional similarities common to components of the “meristematic connectome”. In addition, we present a new model able to explain how portions of the “meristematic connectome” could crosstalk with each other to provide appropriate responses to mechanical stresses.

### 3.1. Morphological and Functional Similarities Present in the “Meristematic Connectome”

All meristematic cells are characterized by a high cell division rate, thin cell wall that is poorly impregnated with lignin or suberin and does not represent a barrier to (in or out) ion diffusion. Additionally, the vacuole is small or divided into ‘sub-vacuoles’. Electron microscopy confirms the occurrence of a very close similarity between the ultrastructural organization of meristematic cells in the RAM and the adjacent PRC [67]. It is interesting that cortical microtubules present in RAM, TZ, and PRC of *Arabidopsis* roots, maintain the same transverse (i.e., perpendicular to the main root axis) orientation while those present in all differentiating tissue change orientation (Figure 5). More recently it has been shown that procambial cells maintain almost all the meristematic properties (i.e., an undifferentiated cellular organization typical of totipotent cells) during the differentiation of vascular bundles [68]. These procambial cells will contribute to VC formation as mentioned above, and for this reason it has been postulated that both the PRC and then VC should be considered as a continuous meristem [69]. VC meristematic cells in roots are more vacuolated as compared to meristematic cells in RAM and PRC [70] but present the same: (i) organelles, (ii) parietal cytoplasm with a rough endoplasmic reticulum (ER), and (iii) microfilaments bundles [71]. In addition, several authors have shown in shoots of conifers (*Abies firma, Abies sachalinensis* and *Larix leptolepis*) [72] and angiosperm trees [72,73], the presence of a complex network of MT and actin filaments in both VC and PRC with higher concentrations in ray cells than in fusiform cells. The need remains to characterize better all the cytological differences existing between fusiform- and/or ray-initials belonging to the same VC which may be the basis of their respective different functional roles in wood production [74].

The similarities between the regulatory mechanisms active in the various components of this “meristematic connectome” are of particular interest [75,76]. In the shoot, homeostasis of both SAM or VC is controlled by a conserved regulatory mechanism involving Class I *knotted*-like Homeobox (*KNOX*) genes (or their paralogs), Class III Homeodomain–leucine zipper and KANADI transcription factors [77]. Groower et al., [78] suggest that stem cells in VC derive from PRC cells, which are in turn derived from stem cells present in SAM.

The literature regarding homologies and differences in homeostasis between RAM and SAM is too wide to be reviewed here. Nevertheless, it is important to highlight the common presence of a group of self-maintaining stem cells forming the quiescent centre (QC) and the organizer centre (OC) respectively in roots and in shoots. Despite the differences, the cytological mechanisms seem to be well conserved with small peptides that move away from the QC or OC to bind distant membrane receptors. This binding activates a cascade of events leading to the synthesis of regulators that control, through a feedback mechanism, the same homeostasis of QC and OC. A similar mechanism seems to be involved in the homeostasis of stem cells in VC and PRC of roots. In fact, there are studies showing how small peptides (CLE41 and CLE44) synthesized in the phloem cells travel (via plasmodesmata) centripetally before being intercepted by the TDR membrane receptor present in the VC cells. This binding affects *WOX4* gene expression involved in VC homeostasis [79]. Regarding PRC homeostasis, Miyashima et al., [80] suggest that the PRC cells belonging to the phloem pole (named protophloem-sieve cells precursors or PSE) produce “inductors” of homeostasis of stem cells present in the same bundle. For this reason, these authors suggest that these cells should be considered as a real “organizer centre” to the same extent as QC and OC. These PSE cells express (under the control of cytokinin) PEAR (Phloem-Early-DOF 1), a transcription factor that moves through the plasmodesmata from cell to cell [80] to maintain stem cell identity. Moreover, it seems that auxin inhibits PEAR synthesis through the HD ZIP III transcription factor to annul homeostasis and induce cell differentiation. However, the findings that homeostasis is present also in PRC cells is important, as these stem cells become (at a later stage) components of the VC [81]. This explains why, VC stem cell homeostasis also involves HD-ZIP III transcription factors. These transcription factors are under auxin control that in turn is controlled by auxin response factor (ARF) and monopteros (ARF5). Nevertheless, it remains to be understood why in the organizer centre present in PRC and VC, this function is assigned only “temporarily” to cells (named “mothers of xylem”) for the time when they remain in a specific position. In addition, at present, it is not clear if another additional organizer centre in PRC and VC also exists in the external position where “mother cells of phloem” are formed. It is interesting that these studies confirm that stem cells in the VC form a unicellular-thick cylindrical layer separating the secondary xylem from the secondary phloem, as first proposed by Sanio in 1873 [82]. 

### 3.2. Crosstalk between Different Components of the “Meristematic Connectome”

The last aspect that we examine in this work concerns the possible crosstalk that may characterize the different components of the “meristematic connectome”. With regard to this, we have built a 3-D model able to examine the longitudinal and transverse distribution of mechanical forces. The 3-D model is based upon a cylindrical structure representing the taproot axis in which the different tissues are represented by 4 concentric layers (Figure 6A). As shown in the cross section we have divided the cylindrical structure by 12 rays which separate adjacent units (theoretically cells) of the same layer (theoretically the same tissue). Applying bending treatment, the model confirms that both the type (i.e., compression with negative values and red coloured, tension with positive values and blue coloured) and magnitude of the mechanical forces changes between adjacent units of same or different layers (from convex to concave side) and along the root axis (ABS, BS and BBS), respectively (Figure 6B,C). Indeed, considering the rays 1 and 7 (concave-convex direction; box in Figure 6B), the model shows that adjacent units of same or opposite tiers are affected by different mechanical forces (box in Figure 6C). Thus, if we assume that one of the layers represents the VC (yellow in Figure 6), in all three bent sectors (ABS, BS and BBS), the VC units placed in the opposite position are differently (in term of magnitude and direction) affected by mechanical forces (Figure 6C).

Although our models do not specifically show the transduction and assessment of mechanical stress signals within VC cells, it helps to better understand that the VC zone is subject to differences in magnitude and type of these forces and is able to activate a differentiated response. An attractive hypothesis is that all the signalling along the root and stem axes takes place within VC initials, but more experiments are necessary to prove this. However, it is reasonable to hypothesize that modifications of cell walls, specific interactions between cell wall and cytoskeleton and alterations of microtubule dynamics [83] may take part in this complex “decision machine”. To the best of our knowledge, there have been very few investigations dealing with the signalling cascade following the detection of environmental stresses taking place in the whole (or a part of) the meristematic connectome. One relevant paper is that by Baluska et al., [84] that suggests that ARK1/STM genes are involved in cell-to-cell signalling in both SAM and VC. Nevertheless, plants perceive and respond to environment signals through several transducers such as: phytohormones, Ca^2+^, electric and/or acoustic signals, and other molecules (reviewed by Leyser [38], and Volkov and Shtessel [85]). In particular, responses of plasma membranes to mechanical signals seem to induce a transient increase of cytosolic Ca^2+^, proton fluxes and ROS generation [86,87,88] despite the fact the regulation of this event remains unclear [89].

Some authors suggest that in *Arabidopsis*, tension forces, acting in the convex side of bent root, induces an increase of Ca^2+^ levels in specific pericycle cells becoming “founder cell” of a new lateral root [53,86]. This Ca^2+^ increase leads to: (a) an alteration in ROS and cytosolic acidification, known to elicit signaling events; (b) a cell wall alkalinization, known to rigidify the cell wall matrix. Diaz-Sala [83] suggested that mechanosensitive ion channels present on the plasma membranes could generate electric action potentials (APs) that propagate on a short distance from cell to cell along with plasma membrane network and through plasmodesmata (or alternatively through phloem cells over a longer distance) inducing modifications of cell walls, specific interactions between cell wall and cytoskeleton, and alterations of microtubule dynamics.

Certainly, a better understanding of the effect of the mechanical stresses upon the whole unit consisting of cytoskeleton-cell wall-plasmalemma [90] could help to understand the communication crosstalk taking place along the different portions of the hypothetical “meristematic connectome”. In particular, we need to investigate the role in signalling played by the MTs of VC initials to understand if they release molecules during the modification of their structure induced by mechanical stress (probably through the involvement of Ca^2+^ movements) [91]. Also, a further in-depth series of studies investigating other type of environmental signals would enhance our understanding of how the meristematic connectome regulates cells’ responses.

In conclusion, this work shows that the VC present along the entire plant body is characterized by an intense crosstalk to coordinate the responses to mechanical stresses, suggesting that this meristem plus the primary and apical meristems could act as single pluricellular interconnected structure, at least in adjusting the plant body architecture to fit better its surrounding environment. The morphological and functional similarities between the meristematic cells, together with the presence of plasmodesmata and occurrence of several conserved mechanisms that control important functions such as homeostasis, supports our proposal to name this structure as the “meristematic connectome”. Thus, the “meristematic connectome” could represent the ideal network enabling a rapid signalling even between distant plant compartments. If this is true then an interesting question arises. Is this functional property acquired ex-novo during VC development or is it inherited by cell lineage (i.e., also conserved in all other meristems such as RAM, SAM, and PRC)? Moreover, it would be interesting to know if other functional activities are conserved in all meristems independently from their position in the plant body. However, the increasing number of reports in recent literature of similarities in the mechanisms controlling important functions such as the homeostasis of RAM, SAM, PRC, and VC, suggests that in future the concept of a presence in plants of a “meristematic connectome” could be extended to include all the sequence of meristems formed during development of the plant body.

## Figures and Tables

**Figure 1 cells-10-02544-f001:**
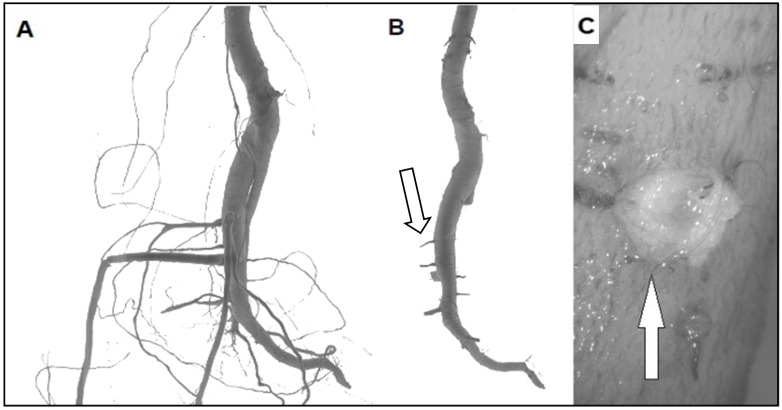
Pruning treatments in *Populus nigra* seedlings. (**A**) Seedlings before pruning showing the LRs. (**B**) Seedlings after pruning treatment with new LRs formation (white arrow). (**C**) New LRs are produced internally to the taproot axis by the VC before protruding from the cork (white arrow).

**Figure 2 cells-10-02544-f002:**
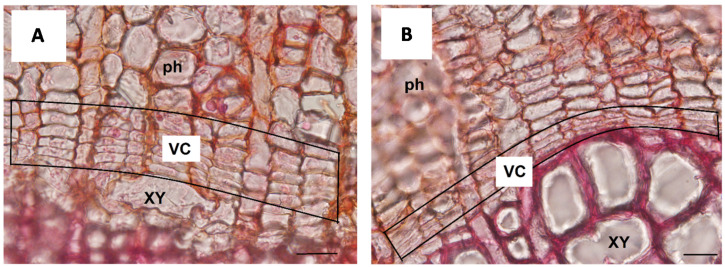
Unidirectional RW production in concave side of Populus nigra bent taproot. (**A**) The VC initials present in the concave side of the bent taproot are characterized by a very high mitotic activity shown by the high number of cells produced. (**B**) The VC initials present in the convex side of the bent taproot are characterized by a reduced mitotic activity shown by the lower number of cells produced XY = xylem; ph = phloem. Scale bars = 20 µm.

**Figure 3 cells-10-02544-f003:**
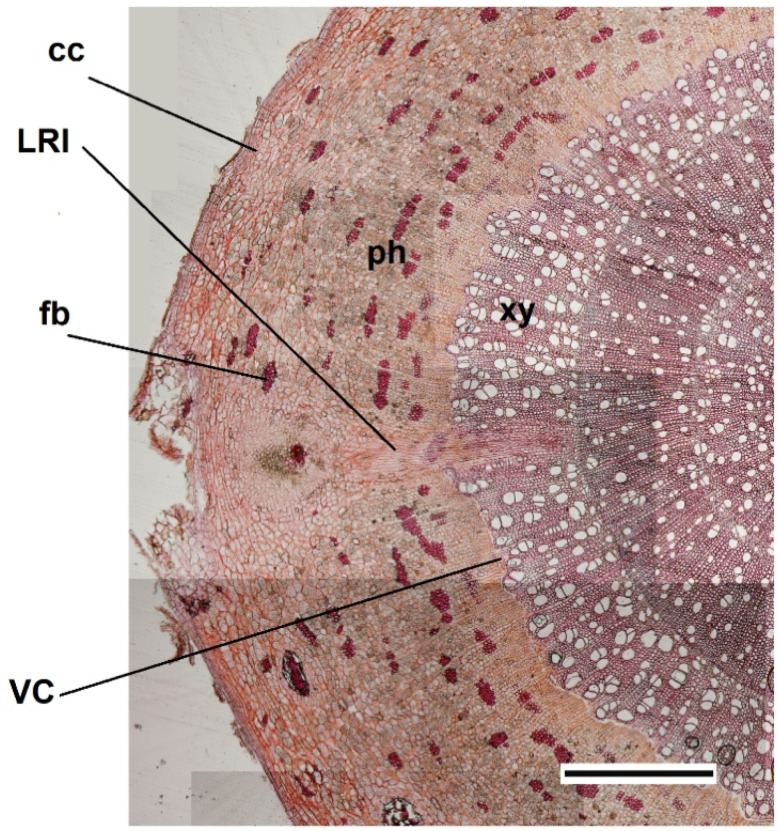
Emergence of a LR produced by the VC initials in the convex side of *Populus nigra* bent taproot. The cross section shows the production of a new lateral root (LR) that is formed by the activity of some VC initials and that grows toward the external convex side of the bent taproot. cc = cortical cells; xy = xylem; ph = phloem; fb = mechanical fibers. Scale bar = 200 µm.

**Figure 4 cells-10-02544-f004:**
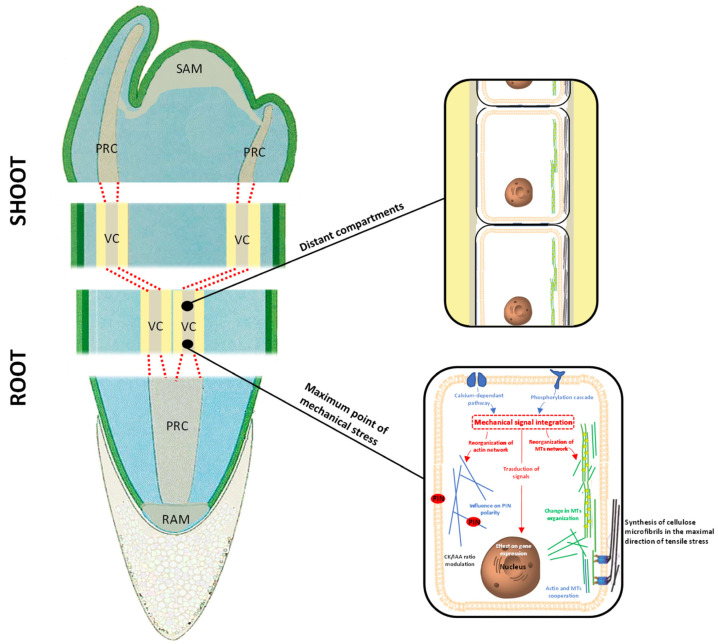
The “meristematic connectome” formed by the sequence of all meristems—RAM, root PRC, root VC, shoot VC, shoot PRC, SAM—represents the ideal network able to induce a rapid signalling between cells/tissues sensing the maximum stress and distant plant compartments.

**Figure 5 cells-10-02544-f005:**
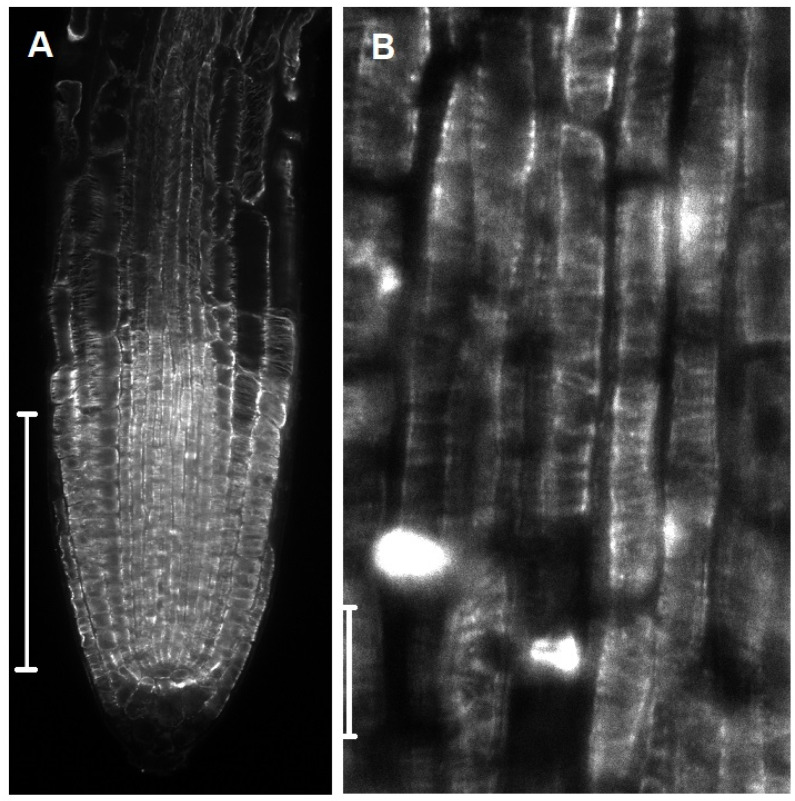
Single confocal laser scanning microscopy (CLSM) sections of *A. thaliana* root tips after α-tubulin immunostaining. (**A**) Central root section revealing a uniform transverse orientation of cortical microtubules in all the developmental zones (RAM, TZ and elongation zones) and all the tissues/cell types. The root zones appear compressed because the root was growing in thick soil (Scale bar = 150 µm). (**B**) Higher magnification of the stele at the border between the meristematic and TZ. All the cell types, including PRC cells, exhibit transverse cortical microtubules (Scale bar = 10 µm).

**Figure 6 cells-10-02544-f006:**
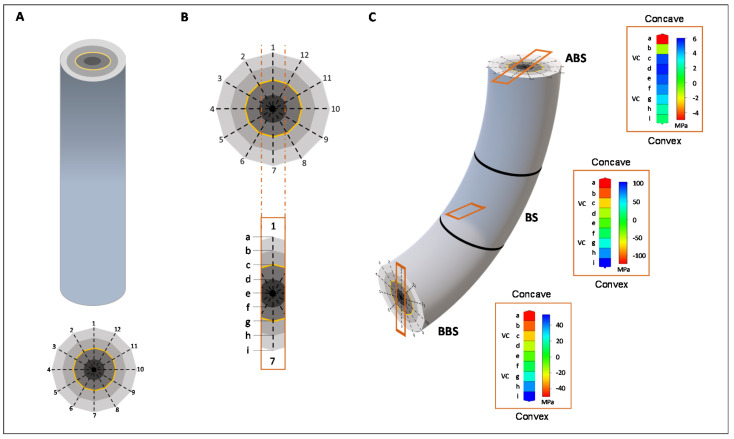
3D-modelling of the longitudinal and transverse mechanical forces distribution. (**A**) Cylindrical structure representing the taproot axis in which the different tissues are represented by four concentric layers. (**B**) In cross-section, 12 rays separate adjacent units (theoretically cell) of the same layer (theoretically same tissue); the continuity between rays 1 and 7 represent the average point concave-convex direction and letters (from a to i) are used to indicate where the force were measured along this direction; (**C**) full-length model of the bent taproot. Each box reports the mean force magnitude of three bent sectors (ABS, BS and BBS) in concave-convex direction (a to i) of a sector flanked by its magnitude scale. In the graphical representations VC is shown as a yellow line in the transverse section while in the boxes the corresponding magnitude of forces is reported.

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
