# Peer review of "Meristematic Connectome: A Cellular Coordinator of Plant Responses to Environmental Signals?"

_cells, 2021, doi:10.3390/cells10102544_

Round 1
Reviewer 1 Report
This is an interesting and thought-provoking manuscript, based on the fascinating hypothesis that all meristematic cells form a structural and functional continuum regulating plant development in response to environmental cues. Such conclusion is reached mainly on the basis of experimental data on the effects of root bending, but the concept is extended to all organs/tissues and to any kind of stress. Specific comments are reported below.
- Some statements appear a bit far-fetched: is there any evidence that "In plants that lack VC (as in some monocots), the “meristematic connectome” could still be active though not involved in secondary growth."? (lines 300-301). About monocots, it is surprising that the authors did not take into consideration those ones that do have secondary growth, and a recent paper on this topic could offer some useful hints (Jura-Morawiec, J., Oskolski, A. & Simpson, P. Revisiting the anatomy of the monocot cambium, a novel meristem. Planta 254, 6 (2021). https://doi.org/10.1007/s00425-021-03654-9).
- Both opening citations (Alpi et al. 2007, Robinson et al. 2020) are papers written in response to other papers in the long (and a bit tiring) debate on plant neurobiology. The authors could find better papers discussing plant ability to coordinate complex interactions, without the academic quarrel.
- The Power of Movement in Plants was authored by both Charles and Francis Darwin (line 44).
- The whole section 2.2 could be written more clearly and concisely.
- The writing is apparently non homogeneous. Some sections are less accurate than others. A non-exhaustive list of suggestions for language improvement: Lines 183-187: related to/with. 184-185: something missing (which resulted?) Line 202 the shoot… responds. Line 241 and 244-245: as a function? Lines 247 and 418 : Maybe transducer is preferable over transductor. Line 328; which may be the basis. Line 384: axis (in… delete the parenthesis. Line 385: shown in (add space). Line 388: both the magnitude and direction change (plural).
Author Response
Dear Reviewer, thank you very much for reading the manuscript and providing comments. Here you can find a list of point-by-point responses to your comments, which were all addressed in the new version of the manuscript. Corrections are tracked. Please note that the number of lines indicated below could not perfectly match with those in the manuscript.
Sincerely yours
Antonio Montagnoli (corresponding author)
- Some statements appear a bit far-fetched: is there any evidence that "In plants that lack VC (as in some monocots), the “meristematic connectome” could still be active though not involved in secondary growth."? (lines 300-301). About monocots, it is surprising that the authors did not take into consideration those ones that do have secondary growth, and a recent paper on this topic could offer some useful hints (Jura-Morawiec, J., Oskolski, A. & Simpson, P. Revisiting the anatomy of the monocot cambium, a novel meristem. Planta254, 6 (2021). https://doi.org/10.1007/s00425-021-03654-9).
Reply: Thanks for this comment. We have reformulated the sentence and included the interesting reference suggested. See lines 332-336.
- Both opening citations (Alpi et al. 2007, Robinson et al. 2020) are papers written in response to other papers in the long (and a bit tiring) debate on plant neurobiology. The authors could find better papers discussing plant ability to coordinate complex interactions, without the academic quarrel.
Reply: We have added other references regarding plant response to environmental signals
- The Power of Movement in Plants was authored by both Charles and Francis Darwin (line 44).
Reply: Thanks for this. Charles has been added (line 44)
- The whole section 2.2 could be written more clearly and concisely.
Reply: The section 2.2 has been shortened and better connected with the presented hypothesis. Also the old figure has been replaced with a summary figure requested by the R#3
- The writing is apparently non homogeneous. Some sections are less accurate than others. A non-exhaustive list of suggestions for language improvement:
Lines 183-187: related to/with.184-185: something missing (which resulted?). Line 202 the shoot… responds. Line 241 and 244-245: as a function? Lines 247 and 418 : Maybe transducer is preferable over transductor. Line 328; which may be the basis. Line 384: axis (in… delete the parenthesis. Line 385: shown in (add space).Line 388: both the magnitude and direction change (plural).
Reply: The English and typing were extensively revised throughout the manuscript. Also, reviewer’s suggestions were included.
Reviewer 2 Report
cells1387625
Meristematic Connectome: A Cellular Coordinator of Plant Responses to Environmental Signals?
The hypothesis presented by the authors is very interesting and merits publication. The data that shows that the VC responds to mechanical stress without the meristem is important. Some English and typos are present.
Comments:
English corrections are needed all over. For example:
Line 75: .. before the "meristematic…
Line 76-77: not clear at all.
Line 81: Start sentence "During the last two …..
Content comments:
Lines 136-148: What are costs and the benefits? A clarification is needed. How is partial Girdling affect bending?
There is no reference in the text for Figure 5.
Figure 6 show that the force on BBS and BS zone are the same? The model does not show what happens in the VC zone.
It was unclear how the section on "Detection and transduction of mechanical stresses" is related to the authors' hypothesis that the structure "meristematic connectome." The Idea that all meristematic cells are connected is appealing. However, in plants except for meristematic cells in the OZ and QZ, all cells are connected via plasmodesmata (as viewed lately in https://doi.org/10.3390/plants10071442).
Author Response
Dear Reviewer, thank you very much for reading the manuscript and providing comments. Here you can find a list of point-by-point responses to your comments, which were all addressed in the new version of the manuscript. Corrections are tracked. Please note that the number of lines indicated below could not perfectly match with those in the manuscript.
Sincerely yours
Antonio Montagnoli (corresponding author)
English corrections are needed all over. For example:
Line 75: .. before the "meristematic…
Line 76-77: not clear at all.
Line 81: Start sentence "During the last two …..
Reply: English has been extensively reviewed throughout the manuscript.
Content comments:
Lines 136-148: What are costs and the benefits? A clarification is needed.
Reply: This part has been revised and a wider explanation is given with few examples. See line 158-167
How is partial Girdling affect bending?
Reply: Girdling did not affect the bending treatment neither the mechanical forces induced since the application of the bending was limited at short or medium term.
There is no reference in the text for Figure 5.
Reply: There was a mistake, Figure 5 is now referenced in the text line 314
Figure 6 show that the force on BBS and BS zone are the same? The model does not show what happens in the VC zone.
Reply: Yes, that is partially true. The models show that the type of forces (tension with positive values and compression with negative values) change across adjacent units (from concave to convex side) while the magnitude of these forces changes along the root axis (ABS from 6 to -4 MPa, BS from 100 to -100 MPa, and BBS from 40 to -40 MPa). And, yes what happen exactly to the VC cells is not shown here. We have rephrased this sentence and incorporated the reviewer’s comment. See lines 441-469
It was unclear how the section on "Detection and transduction of mechanical stresses" is related to the authors' hypothesis that the structure "meristematic connectome." The Idea that all meristematic cells are connected is appealing. However, in plants except for meristematic cells in the OZ and QZ, all cells are connected via plasmodesmata (as viewed lately in https://doi.org/10.3390/plants10071442).
Reply: The section 2.2 has been shortened and better connected with the presented hypothesis. A figure has been removed since it did not add much to what is reported in the text. However, according to the R#3 we added in this section a summary figure. The plasmodesmata transmission of virus has been added with the related references
Reviewer 3 Report
In the study 'Meristematic Connectome: A Cellular Coordinator of Plant Re
sponses to Environmental Signals?', Chiatante propose an interconnected structure 'meristematic connectome' that serves as the regulatory network enabling a rapid signalling even between distant plant body compartments. The authors proposed this hypothese through comprehensive literature review and their own research work. The hypothesis is novel, provocative and well grounded. Some minor revisions are recommended.
1) abbreviations only needs to be mentioned once.
2) The authors should make a summery figure of proposed meristematic connectome and how it regulates cells' response to environmental signals.
3) The mechanical stresses are the main environmental signals mentioned in this review. Other environmental signals were not mentioned by much. The authors should also make some effort to distinguish different types of environmental signals and discuss how meristematic connectome regulates cells' response to them.
4)what's the significance of Fig.5? It is not mentioned in the text.
Author Response
Dear Reviewer, thank you very much for reading the manuscript and providing comments. Here you can find a list of point-by-point responses to your comments, which were all addressed in the new version of the manuscript. Corrections are tracked. Please note that the number of lines indicated below could not perfectly match with those in the manuscript.
Sincerely yours
- abbreviations only needs to be mentioned once.
Reply: Authors have double checked the abbreviation throughout the text and now are mentioned only in the Abstract (since this section need to be self-standing) and in the very first part of the Introduction (page 2 and 3)
- The authors should make a summery figure of proposed meristematic connectome and how it regulates cells' response to environmental signals.
Reply: A summary figure has been added in the section 2.2
- The mechanical stresses are the main environmental signals mentioned in this review. Other environmental signals were not mentioned by much. The authors should also make some effort to distinguish different types of environmental signals and discuss how meristematic connectome regulates cells' response to them.
Reply: The study presented here focused on mechanical forces since the authors have collected in-depth data (i.e., molecular, anatomical, and morphological response) especially concerning this type of environmental stress that might originate from the wind or from a steep slope. We added this to the concluding section of the manuscript (577-579).
- what's the significance of Fig.5? It is not mentioned in the text.
Reply: Apologize for this mistake. Figure 5 is now referenced in the text line 427